# Genomic Insights into Tibetan Sheep Adaptation to Different Altitude Environments

**DOI:** 10.3390/ijms252212394

**Published:** 2024-11-19

**Authors:** Wentao Zhang, Chao Yuan, Xuejiao An, Tingting Guo, Caihong Wei, Zengkui Lu, Jianbin Liu

**Affiliations:** 1Key Laboratory of Animal Genetics and Breeding on Tibetan Plateau, Ministry of Agriculture and Rural Affairs, Lanzhou Institute of Husbandry and Pharmaceutical Sciences, Chinese Academy of Agricultural Sciences, Lanzhou 730050, China; m18251871965@163.com (W.Z.); yuanchao@caas.cn (C.Y.); anxuejiao@caas.cn (X.A.); guotingting@caas.cn (T.G.); 2Sheep Breeding Engineering Technology Research Center of Chinese Academy of Agricultural Sciences, Lanzhou 730050, China; 3State Key Laboratory of Animal Biotech Breeding, Institute of Animal Sciences, Chinese Academy of Agricultural Sciences, Beijing 100193, China; weicaihong@caas.cn

**Keywords:** Tibetan sheep, whole-genome resequencing, selection signature, high altitude, ultra-high altitude

## Abstract

In recent years, research has gradually uncovered the mechanisms of animal adaptation to hypoxic conditions in different altitude environments, particularly at the genomic level. However, past genomic studies on high-altitude adaptation have often not delved deeply into the differences between varying altitude levels. This study conducted whole-genome sequencing on 60 Tibetan sheep (Medium Altitude Group (MA): 20 Tao sheep (TS) at 2887 m, High Altitude Group (HA): 20 OuLa sheep (OL) at 3501 m, and Ultra-High Altitude Group (UA): 20 AWang sheep (AW) at 4643 m) from different regions of the Tibetan Plateau in China to assess their responses under varying conditions. Population genetic structure analysis revealed that the three groups are genetically independent, but the TS and OL groups have experienced gene flow with other northern Chinese sheep due to geographical factors. Selection signal analysis identified *FGF10*, *MMP14*, *SLC25A51*, *NDUFB8*, *ALAS1*, *PRMT1*, *PRMT5*, and *HIF1AN* as genes associated with ultra-high-altitude hypoxia adaptation, while *HMOX2*, *SEMA4G*, *SLC16A2*, *SLC22A17*, and *BCL2L2* were linked to high-altitude hypoxia adaptation. Functional analysis showed that ultra-high-altitude adaptation genes tend to influence physiological mechanisms directly affecting oxygen uptake, such as lung development, angiogenesis, and red blood cell formation. In contrast, high-altitude adaptation genes are more inclined to regulate mitochondrial DNA replication, iron homeostasis, and calcium signaling pathways to maintain cellular function. Additionally, the functions of shared genes further support the adaptive capacity of Tibetan sheep across a broad geographic range, indicating that these genes offer significant selective advantages in coping with oxygen scarcity. In summary, this study not only reveals the genetic basis of Tibetan sheep adaptation to different altitudinal conditions but also highlights the differences in gene regulation between ultra-high- and high-altitude adaptations. These findings offer new insights into the adaptive evolution of animals in extreme environments and provide a reference for exploring adaptation mechanisms in other species under hypoxic conditions.

## 1. Introduction

The Tibetan Plateau, often referred to as the “Roof of the World”, is the highest and most extensive plateau on Earth, with an average elevation exceeding 4000 m above sea level. This harsh environment, characterized by low oxygen levels, intense UV radiation, and cold temperatures [1,2,3], provides an exceptional setting for studying high-altitude adaptation. Among the species that have successfully adapted to these harsh conditions, Tibetan sheep (Ovis aries) offer a particularly valuable case for examining the genetic basis of altitude adaptation [4]. Tibetan sheep, having inhabited the plateau for thousands of years, provide a valuable opportunity to explore the genetic basis of high-altitude adaptation and the evolutionary processes that have enabled their survival in such extreme environments.

Research on altitude adaptation has revealed several key genes associated with hypoxia tolerance, metabolic adjustments, and other physiological changes crucial for survival at high altitudes. Notably, genes such as *EPAS1*, *EGLN1*, and *HIF1A* have been identified in both humans and animals as being central to high-altitude adaptation. *EPAS1* [5,6] regulates the body’s response to low oxygen levels by modulating erythropoiesis. Under hypoxic conditions, *EGLN1* [7,8,9] aids high-altitude adaptation and survival by stabilizing HIF proteins. *HIF1A* [10,11] acts as a major regulator of oxygen homeostasis, and it enhances oxygen delivery and metabolism by activating various target genes involved in angiogenesis and erythropoiesis under hypoxic conditions. Beyond these well-known genes, research has identified several other genetic adaptations linked to high-altitude survival. For example, *PRKAA1* [10], which encodes the alpha 1 catalytic subunit of AMP-activated protein kinase (AMPK), plays a role in energy metabolism and has been implicated in the regulation of glucose and lipid metabolism at high altitudes. Additionally, recent studies have also highlighted the role of mitochondrial [12,13,14] DNA variations in high-altitude adaptation.

The study of selection signals has played a key role in unraveling the genetic basis of high-altitude adaptation. Advances in next-generation sequencing technologies and computational tools (including alignment tools, variant calling tools, and population genetics tools) have enabled more refined genome-wide analyses [15]. These modern techniques provide higher resolution and a greater ability to detect subtle selection signals, allowing researchers to more precisely identify adaptive loci.

In this study, we performed whole-genome resequencing of Tibetan sheep populations sampled from multiple altitudes on the Tibetan Plateau. These samples were collected from a relatively low-middle altitude (around 3000 m) in the northeastern part of the Tibetan Plateau, as well as from a high altitude (around 3500 m) and an ultra-high altitude (around 4500 m). Genome-wide scans were then performed using three selection signal analysis methods to identify specific genetic signals that enable these Tibetan sheep to adapt to harsh environments. This study deepens our understanding of the genetic mechanisms of high-altitude adaptation and contributes to the broader field of evolutionary biology.

## 2. Results

### 2.1. Whole-Genome Sequencing and Genetic Variation

After quality control, a total of 988,175,526,652 bp of high-quality clean data were obtained (corresponding to 6,859,155,228 high-quality clean reads) (see Appendix A). The average total mapping ratio for the Tibetan sheep reached 99.25%, with an average sequencing depth of 6.32× (see Appendix A). Through variant annotation, a total of 29,722,880 SNPs were identified (see Figure 1). The TS/TV (transition/transversion) ratio was determined to be 2.02, indicating a standardized genomic population structure. The occurrence of nonsynonymous mutations is more frequent than that of synonymous mutations.

### 2.2. Population Genetic Analysis

We conducted population genetic analysis of three groups (AW: AWang sheep, OL: OuLa sheep, and TS: Tao sheep) using high-quality SNP data to understand the genetic relationships and differences between these groups. First, the PCA (principal component analysis) plot (Figure 2a) shows that PC1, PC2, and PC3 explain 4.23%, 3.52%, and 2.93% of the genetic variation, respectively. The plot illustrates the clustering of the three groups, with their distribution in genetic space being relatively close; among them, the OL group is more concentrated, while the AW and TS groups form distinct clusters. Secondly, cross-validation error (Figure 2d) suggests that K = 2 might be the best modeling choice. Ancestral component analysis (Figure 2b) reveals that when K = 2, the AW group is predominantly represented by the red component, whereas the OL and TS groups show a noticeable increase in the blue component. Lastly, the neighbor-joining tree analysis (Figure 2c) clearly depicts the phylogenetic relationships among the three groups, which is consistent with the PCA and ancestry component analysis results.

### 2.3. Analysis of Selection Signals

The three selection signal analysis methods (Figure 3a–c) identified 1403, 1783, and 1243 significant SNPs in the comparisons of AW-OL, AW-TS, and OL-TS, respectively. Annotation of these SNPs yielded 394, 478, and 364 candidate genes, respectively (Figure 3d; Appendix A). Among these, there are 158 common candidate genes between the comparisons of AW-OL and AW-TS groups, 10 common candidate genes between the comparisons of AW-OL and OL-TS groups, and 74 common candidate genes between the comparisons of AW-TS and OL-TS groups. Additionally, a total of 7 genes were present in all three comparison groups.

### 2.4. Analysis of GO and KEGG Enrichment

Functional analysis (Figure 4a; Appendix A) of ultra-high-altitude adaptation candidate genes reveals their involvement in processes related to lung development (e.g., alveolar sac development, type II alveolar cell differentiation), angiogenesis (VEGF signaling), metabolic regulation (autophagy, TORC1), neural adaptation, oxidative stress, and hemoglobin synthesis. These biological processes collectively assist cells in surviving, developing, and maintaining metabolic balance under hypoxic conditions.

Functional analysis (Figure 4b; Appendix A) of high-altitude adaptation candidate genes indicates their involvement in cellular responses to hypoxia, iron homeostasis, mitochondrial DNA replication, glycogen synthesis and metabolic regulation, calcium signaling pathways, protein ubiquitination, and histone modification. These processes directly or indirectly participate in how cells regulate energy metabolism, oxygen transport, calcium balance, angiogenesis, and gene expression under hypoxic conditions.

Moreover, shared candidate genes across multiple comparison groups are enriched in processes (Figure 4c; Appendix A) such as fatty acid degradation, adipocytokine signaling pathways, peroxisome function, long-chain and very long-chain fatty acid metabolism, and acyl-CoA metabolism. Notably, the HIF-1 signaling pathway, ferroptosis, PPAR signaling pathway, and VEGF signaling pathway play crucial roles, as they are central to cellular metabolism and survival mechanisms in low-oxygen environments.

## 3. Discussion

A sequencing depth of 6.32× significantly improves the reliability of variant annotation, mutation detection, and genomic structure exploration. The sequencing achieved a high matching rate of 99.25%, indicating thorough coverage of the reference genome and ensuring precise genomic analysis. The TS/TV ratio close to 2 suggests a well-balanced genome with a preserved structure [16], which supports the reliability of data for studies on population structure and selection signals. The more frequent occurrence of nonsynonymous mutations suggests that under altitude pressure, the Tibetan sheep genome has undergone positive selection, with mutations more likely to alter protein function. This study provides dependable data for Tibetan sheep genomic research, contributing to population genetic evolution studies.

Low explained variance from the PCA analysis can often indicate that the genetic background of the samples is complex. This complexity may stem from the presence of mixed or subpopulations within the data or from the influence of various selective pressures, making it challenging to capture group differences through a few principal components. However, in our study, PC1–3 have already clearly separated the different groups, indicating that despite the low variance explained by individual components, the main population structures are sufficiently captured by the first few principal components. The PCA and NJ tree results indicate that there is a certain degree of genetic differentiation among the populations, with each being genetically independent. Although it was expected that the TS and OL groups would have a higher proportion of the red component (the high-proportion ancestral component), the actual analysis showed that the proportion of the red component in the TS and OL groups is lower than that in the AW group. Geographic analysis reveals that the AW group is located in the rugged Hengduan Mountains in the eastern Tibetan Plateau, which is geographically isolated from other sheep populations. This geographic isolation may have led to the AW group retaining a relatively unique red component genetically. In contrast, the TS and OL groups inhabit the relatively flat mid-altitude region in the northeastern Tibetan Plateau, which has more opportunities for gene flow with northern Chinese sheep populations. Therefore, the proportion of the red component in the TS and OL groups may be influenced by gene flow and admixture with other populations, resulting in a lower proportion of the red component. These geographic and environmental differences may be key reasons for the unexpected distribution of genetic components.

Adaptation to altitude in animals primarily involves adaptation to hypoxic environments, resistance to high ultraviolet radiation, and energy utilization [17,18]. This study focuses on hypoxic adaptation, which includes improved blood oxygen transport capacity, regulation of hypoxia response pathways, oxidative stress protection mechanisms, metabolic adaptation, and enhanced cardiopulmonary function. In analyzing the functions of genes associated with ultra-high and high-altitude adaptation, we observe both differences and commonalities in their adaptation mechanisms. Both ultra-high- and high-altitude adaptation genes involve mechanisms for responding to and adapting to low oxygen, such as cellular responses to hypoxia, VEGF signaling pathways, and hemoglobin synthesis. Additionally, metabolic adaptation, including glycogen synthesis and energy metabolism regulation, is a common feature of both. However, there are differences in their adaptation mechanisms: ultra-high-altitude adaptation genes focus more on mechanisms that directly affect oxygen acquisition, such as lung development, angiogenesis, and red blood cell production, while high-altitude adaptation genes are more concerned with internal cellular regulation mechanisms, such as mitochondrial DNA replication, iron homeostasis, and calcium signaling pathways, which help cells maintain normal function under prolonged hypoxic conditions.

Unique candidate genes for ultra-high-altitude adaptation are primarily associated with angiogenesis, oxygen utilization, oxygen uptake, oxidative stress resistance, erythropoiesis, and cellular stress responses. *FGF10* and *MMP14* play roles in angiogenesis and tissue repair [19,20]. *SLC25A51* and *NDUFB8* are crucial for mitochondrial redox reactions and the respiratory chain, affecting the efficiency of intracellular oxygen utilization [21,22]. *ALAS1* is a rate-limiting enzyme in heme biosynthesis; heme is a core component of hemoglobin, which is essential for oxygen uptake and transport [23]. *PRMT1* and *PRMT5* are necessary for erythropoiesis, jointly regulating post-translational processing and the formation of red blood cells [24,25]. *HIF1AN* regulates the HIF-1 signaling pathway, playing a role in cellular stress and hypoxic responses. These genes are related to different mechanisms of altitude adaptation and may help organisms adapt to ultra-high-altitude hypoxic environments by regulating angiogenesis, oxygen utilization, erythropoiesis, and oxidative stress.

Candidate genes for high-altitude adaptation differ significantly from those for ultra-high-altitude adaptation. *HMOX2* is involved in the degradation of heme, producing CO, which is important for regulating vascular relaxation and oxidative stress, thus affecting oxygen utilization [26]. Additionally, *HMOX2* plays a key role in regulating angiogenesis, particularly in maintaining endothelial cell function and vascular stability [27]. The Semaphorin family is involved in angiogenesis [28,29]; thus, the high-selection gene *SEMA4G* is also speculated to play a role in angiogenesis under hypoxic conditions. *SLC16A2* affects lactate metabolism by regulating thyroid hormone levels, thereby enhancing oxygen utilization [30]. The role of *SLC22A17* in iron homeostasis is indirectly related to erythropoiesis [31]. The *BCL2L2* gene prevents apoptosis due to oxidative damage or hypoxia under low-oxygen stress conditions [32]. These genes contribute to various adaptive mechanisms, helping the body cope with challenges in high-altitude environments by regulating these mechanisms.

Shared genes are crucial core genes in altitude adaptation, playing significant roles in both high-altitude and ultra-high-altitude environments. The *ERG* gene is essential for vascular development and angiogenesis [33]. Variants in the *HBE1* and *HBE2* genes may affect fetal hemoglobin expression, influencing individual adaptation to hypoxic conditions. *HBE1* and *HBE2* genes function by regulating the oxygen affinity of fetal hemoglobin, affecting hemoglobin composition and supporting fetal adaptation in low-oxygen environments. *HMGN1* may influence cellular responses to oxidative stress by regulating chromatin structure. *ACSL6* is involved in lipid metabolism under hypoxic stress [34], suggesting that adaptation to low oxygen might rely on increased fatty acid oxidation. *PIK3R3* can activate HIF-1 and PIK/Akt pathways to respond to hypoxic stress, including angiogenesis and erythropoiesis [35,36]. Among the mutation sites of these genes, some sites show a significant gradient change in mutation frequency with increasing altitude (Appendix A). This suggests that these genes may play a key regulatory role in the adaptation process to different altitudinal environments, helping the species cope with the specific environmental pressures at each altitude and promoting adaptive evolution. The functions and enrichment results of these genes indicate that lipid metabolism, cellular metabolism, cellular function regulation, and chromatin regulation play important roles in altitude adaptation. These mechanisms involve energy regulation, cell survival, metabolic adaptation, and gene regulation, which help organisms maintain physiological function and adaptability in extreme-altitude environments.

Understanding the differential gene regulation for adaptation to ultra-high and high altitudes provides an opportunity to tailor breeding strategies that better align with specific environmental challenges. For example, ultra-high-altitude adaptation focuses on direct mechanisms for oxygen intake, while high-altitude adaptation regulates the sustained function of cells under low-oxygen conditions. This distinction offers precise guidance for breeding animals under different environmental pressures. Meanwhile, gradient mutation sites can serve as markers for altitudinal adaptation strength, enabling the assessment of a breed’s ability to adapt to specific altitudes. By screening these sites, breeding strategies can be optimized to select livestock breeds with enhanced environmental adaptability.

## 4. Materials and Methods

### 4.1. Ethics Statement

The experimental protocols involving sheep were approved by the Animal Ethics Committee at the Lanzhou Institute of Husbandry and Pharmaceutical Sciences, Chinese Academy of Agricultural Sciences (NO. 20231447).

### 4.2. Sample Collection and Resequencing

The sample consisted of sixty adult grazing healthy rams (20 Tao sheep (TS) at 2887 m, 20 OuLa sheep (OL) at 3501 m, and 20 AWang sheep (AW) at 4643 m) with pure white coats and no spots. The Fis (Fixation Index within Subpopulation) values are in Appendix A. The groups selected in this study all originate from various purebred conservation areas, where some inbreeding has occurred within each population. Blood was collected from the jugular vein of the sheep. Genomic DNA was isolated utilizing the TIANamp Blood DNA Kit (Tian Gen Biotech Co., Ltd., Beijing, China), with its purity and concentration (Appendix A) assessed using a Nanodrop 2000 spectrophotometer (Thermo Scientific, Wilmington, NC, USA). The extracted DNA was enzymatically fragmented, end-repaired, and A-tailed. Sequencing adapters were ligated to the DNA fragments, which were then purified using AMPure XP beads. DNA fragments ranging from 300 to 400 bp were selected and subjected to PCR amplification. The resulting library underwent further purification and quality control before being sequenced on a Hiseq X10 PE150 platform. The raw sequencing data were saved in FASTQ format for subsequent analysis.

### 4.3. Quality Control and Alignment

The filtering process, outlined in [15], included the following steps: (1) discarding reads containing adapter sequences, (2) eliminating reads with an N ratio exceeding 10%, and (3) filtering out low-quality reads where more than 50% of bases had a quality score of Q ≤ 20. High-quality reads were then aligned to the self-assembled genome_HB using BWA (version 0.7.15) [37] with the MEM algorithm (parameters: −k 32 −M). The resulting SAM files were converted to the BAM format using SAMtools, and duplicate reads were marked using Picard (version 2.18.7). Coverage statistics were generated using bedtools (v2.25.0) [38]. Variants were annotated with ANNOVAR [39], and SNPs were further filtered by excluding loci with a missing data rate above 20% and a minor allele frequency (MAF) below 5% [40,41].

### 4.4. Population Structure Analysis

SNPs were pruned using PLINK 1.09’s [42] indep-pairwise [43] command with the following parameters: a 25-SNP window, a 5-SNP step size, and an r^2^ threshold of 0.05. Principal component analysis (PCA) was conducted using PLINK 1.09 to discern genetic clusters. A neighbor-joining tree [43] was generated using treebest [44] and subsequently visualized using ITOL [45] (https://itol.embl.de/upload.cgi, accessed on 30 July 2024). The population genetic structure [46] was examined with admixture [47]. The best K value for modeling was determined through k-fold cross-validation [48].

### 4.5. Selection Signal Analyses

This study utilized three approaches to analyze selection: the Pairwise Fixation Index (F_ST_) [49], π ratio [50,51], and Tajima’s D [52]. The integration of F_ST_ and π ratio analyses provides a comprehensive view of genetic variation, facilitating the detection of selection signals and evolutionary patterns in populations adapting to various altitude environments. Tajima’s D was then applied to further evaluate whether the identified loci were evolving neutrally or under selective pressure. These analyses were performed on filtered SNP data using the PopGenome software [53,54], applying sliding windows of 100 kb with 10 kb steps [15]. Data visualization was accomplished through custom R scripts [55].

### 4.6. Screening and Functional Analysis of Candidate Genes

We then focused on identifying potential selection signals associated with altitude adaptation. Candidate signals were identified by selecting the top 5% of overlapping SNP loci based on F_ST_ and π ratio values. These loci were further evaluated for genetic differentiation using Tajima’s D. Gene annotation was conducted through ANNOVAR, followed by Gene Ontology (GO) and Kyoto Encyclopedia of Genes and Genomes (KEGG) enrichment analyses to explore the regulatory mechanisms underlying altitude adaptation. Functional categorization of candidate genes was performed using DAVID 6.8 [56] (https://david.ncifcrf.gov/, accessed on 30 July 2024), and pathway enrichment analysis was conducted with Kobas 3.0 [57] (http://kobas.cbi.pku.edu.cn/kobas3/genelist/, accessed on 30 July 2024). Enriched GO and KEGG pathways were visualized using Bioinformatics tools [58] (https://www.bioinformatics.com.co.uk, accessed on 10 July 2024).

## 5. Conclusions

Research shows that mechanisms of hypoxic adaptation vary with altitude. Commonalities include responses to low oxygen and metabolic regulation. However, at ultra-high altitudes, genes focus more on promoting lung development, angiogenesis, and erythropoiesis, which directly affect oxygen intake. In contrast, high-altitude adaptation genes are more concerned with internal cellular regulation mechanisms, such as mitochondrial DNA replication, iron homeostasis, and calcium signaling pathways, which help cells maintain normal function under prolonged hypoxic conditions. In these processes, some shared genes (*ERG*, *HBE1*, *HBE2*, *HMGN1*, *ACSL6*, and *PIK3R3*) play a crucial universal role in adaptation across different altitudes. Despite differences in adaptation mechanisms at varying altitudes, these core shared genes provide Tibetan sheep with survival advantages in different low-oxygen environments.

## Figures and Tables

**Figure 1 ijms-25-12394-f001:**
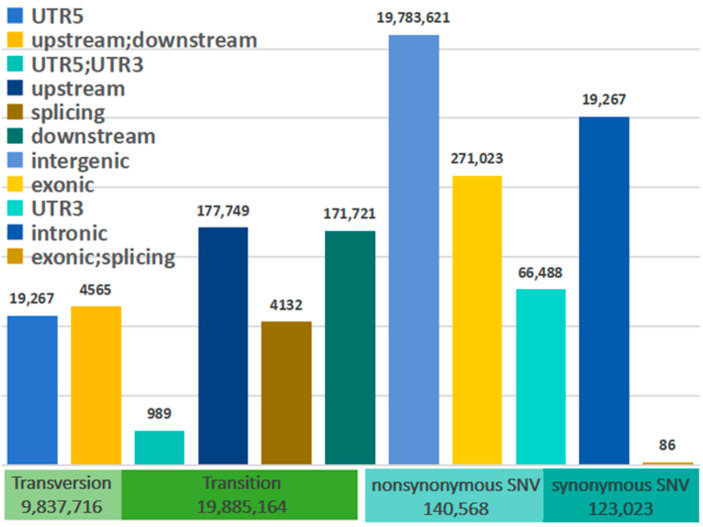
The types of SNP variants in the whole-genome region. Note: “Upstream; downstream” refers to the 1 kb region upstream of one gene and also the 1 kb region downstream of another gene. “UTR5; UTR3” means that it contains the start region of one gene and the termination region of the previous gene.

**Figure 2 ijms-25-12394-f002:**
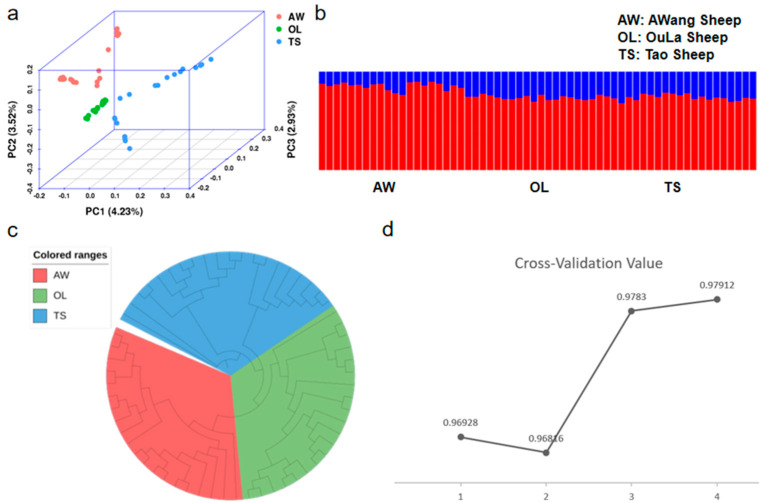
Analysis of the genetic structure of the population: (**a**) principal component analysis (PCA) (each color represents a sheep population); (**b**) K = 2, population structure analysis (each color represents a specific ancestral component); (**c**) individual evolutionary tree (NJ tree) (each color represents a sheep population); and (**d**) cross-validation error (X axis represents K value).

**Figure 3 ijms-25-12394-f003:**
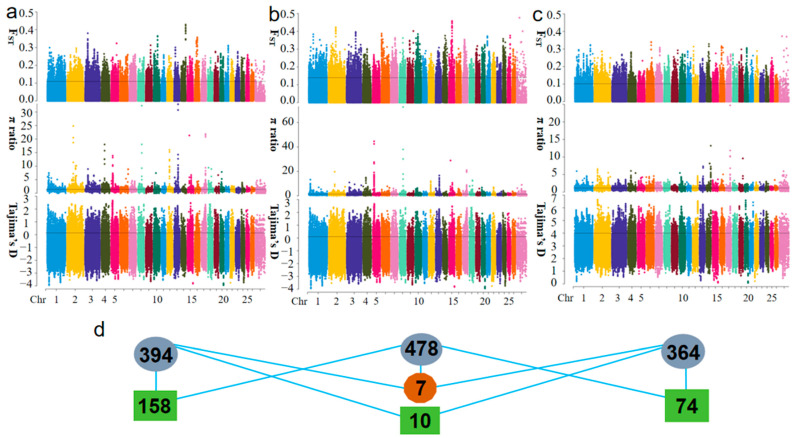
Selection signal analysis: (**a**) Fst, π ratio, Tajima’s D selection elimination (AW-OL); (**b**) Fst, π ratio, Tajima’s D selection elimination (AW-TS); (**c**) Fst, π ratio, Tajima’s D selection elimination (OL-TS); and (**d**) the intersection among the comparison groups.

**Figure 4 ijms-25-12394-f004:**
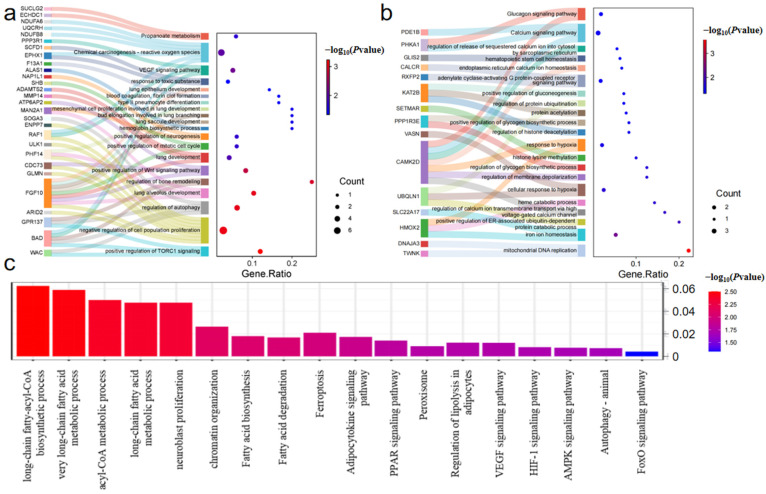
GO enrichment and KEGG enrichment results: (**a**) Sankey diagram of ultra-high-altitude adaptation candidate genes; (**b**) Sankey diagram of high altitude adaptation candidate genes; and (**c**) bar chart of shared genes.

## Data Availability

The datasets generated and analyzed in the current project (PRJNA1138910) are deposited in the NCBI SRA repository (http://www.ncbi.nlm.nih.gov/bioproject/1138910 (accessed on 1 September 2024)).

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
