# Peer review of "Genomic Insights into Tibetan Sheep Adaptation to Different Altitude Environments"

_ijms, 2024, doi:10.3390/ijms252212394_

Round 1

Reviewer 1 Report

Comments and Suggestions for Authors

The authors provide a Genomic Insights into Tibetan Sheep Adaptation to Different Altitude Environments, reporting valuable novel information about adaption of such breeds to harsh environments. There are some missing or unlinked points that to my opinion the authors should elaborate more giving a better explanation assisting the improvement of there work. How different sequencial information (i.e. mutations) is linked with altidute difference? Are thes mutations lethal? Which are the border lines for distingushing high or very high altitude environments?  Sharing the same genes of course support a common biochemical mechanism but how this mechanisms are chenged due to alltitude? Are there any differences in the observed common genes between the tested breeds? Further to these points the authors could improve their article considering the following points:

l. 84 specify the abbreviations TS/TV

Figure 1: Authors report various categories of identified sequences however is not so clear in some cases. In example they refer upstream/down stream and then seperate downstream upstream , or UTR5/UTR3 and then seperate UTR3 UTR5 which is not the sum of the two ones. Could you explain and elaborate more on the figure?

l 90-92 Report the KMO index for the PCA analysis. The low score of genetic variability explained by the three Principal components may suggest that PCA is not the proper analysis for clustering. How can you interpret the low percentange of genetic variab;ility explzined by the three PCs?

l.96 how cros validation error was estimate? Report in materials.

l. 146-148 & 168-169 support with reference

l 174-184 support with the evidence retreived by your study.

l. 174 -176 what does it mean high-altitude and ultra -high altitude adaptation mechanisms? Of course altitude influences adaptation mechanism but which is the border line for distingushing between them? specify and explain in the introduction section and also link with the habitat in terms of altituted of the used breeds.

l 211-226. Common genes are essential for biochemical purposes, however also differential mutations could support mechanisms under different environments. Did you notice any point mutations in these common genes between the analyzed breeds or the reference genome that can be link with the differenr altittude and explain more the adaptation mechanisms? Please elaborate more.

Also authors should add a section in the discussion how can these different adaptation mechanisms, genes screeining and sequence analysis could support the future of the animal genetics and animal husbandry.

l.233-234 a verb is missing

l. 233 dis authors assure that the animals of each breed were not relatives? how? please specify in the text

l. 237-238 specify the initial DNA concentration used 

Reviewer 2 Report

Comments and Suggestions for Authors

The introduction is sufficiently informative, yet concise. The references used are new and adequate. The experimental protocols were approved by the Animal Ethics Committee. The processing methodology is appropriate. Presentation of results is appropriate.

I recommend publishing the manuscript after taking into account my comments below:

In Chapter 4.2. I recommend specifying the pedigree-based average kinship between the selected animals. This would be important so that the Reader knows how representative the sampling is. Alternatively, the authors provide the molecular Fis values for the three breeds.

The tables in the supplementary material are useful. In the header of these, I recommend indicating that the decimal point is a comma; or improve (e.g. Table S4.).

Regarding Figure 1, I would like to note that the explanation of the abbreviations must be given as a continuation of the title of the figure. All the more so because they are also missing from the text. It is a general requirement that the figures can be interpreted independently!

The situation is similar with Figure 2. The abbreviations of the breed names are not given clearly anywhere in the manuscript, this can only be inferred. I recommend that this be made clear in Chapter 2.2 (and additionally in Figure 2), where the results by breed appear for the first time in the manuscript. Furthermore, in the explanation of Figure 2, the meaning of the colors should be indicated separately. Third, the order of the sections in Chapter 2.2 and Figure 2 must match. It's not now. I recommend following the order a-b-c-d in the text as well. In Figure 2 title, N-J is to change for NJ.

In chapter 2.3, I recommend reconsidering the use of "vs". Because here we are not contrasting two groups, but comparing two groups (comparison of AW and OL etc.).

Figure 3 is missing which indicator the values on the y-axis refer to. One can only suspect what the Fst, π ratio and Tajima's D values might be. The choice of three parts of the figure should be considered, because they are too small, and the values are only displayed graphically. I consider it necessary to present the results of the statistical test, e.g. in which Fst and D values did the Authors find a significant difference?

Round 2

Reviewer 1 Report

Comments and Suggestions for Authors

The authors addressed the majority of my comments, incorporating them in the revised version that they submitted.

I have some minor points in regard to their response to  some of my comments and the relative text in the manuscript, which is as follows:

a) Response 3 (Relative to Comment 3) in regard to PCA analysis. Since the proportion of variance explained by the principal components in the PCA plot tends to be relatively low, as authors claim, I suggest that KMO score should be quite low <0.5. This, statistically makes the analysis inappropriate for clustering and explaining any type of different clusters. Therefore the authors should clearly highlight the limitation of this approach or omit this type of presentation or use other type of statistical approach for clustering (i.e. MDA).

b) Response 11 (Relative to comment 11) regarding inbreeding. Inbreeding may be used in conservation programs to retain purity of the breed. However, as a process it may favor some genetic loci contrary to other loci. How sure are you by claiming that "inbreeding does not impact the main objective of this study, which is  to observe the adaptive evolution of the populations during their migration process"? Could the inbreeding value together with the rearing environment and productivity support this?
